# Test of Gross Motor Development-3: Item Difficulty and Item Differential Functioning by Gender and Age with Rasch Analysis

**DOI:** 10.3390/ijerph19148667

**Published:** 2022-07-16

**Authors:** Nadia Cristina Valentini, Marcelo Gonçalves Duarte, Larissa Wagner Zanella, Glauber Carvalho Nobre

**Affiliations:** 1Department of Physical Education, Physiotherapy and Dance, Universidade Federal do Rio Grande do Sul, Porto Alegre 90690-200, Brazil; nadia.cristina@ufrgs.br (N.C.V.); duarte.marcelo@ufms.br (M.G.D.); larissa.zanella@sertao.ifrs.edu.br (L.W.Z.); 2Department of Physical Education, Universidade Federal do Mato Grosso do Sul, Campo Grande 79070-900, Brazil; 3Department of Sports and Leisure, Instituto Federal de Educação, Ciência e Tecnologia do Rio Grande Do Sul, Sertão 99170-000, Brazil; 4Departamento de Educação Física, Instituto Federal de Educação, Ciência e Tecnologia do Ceará, Fortaleza 60040-531, Brazil

**Keywords:** TGMD-3, child development, assessment, motor development, Rasch analysis, fundamental motor skills

## Abstract

The assessment of motor proficiency is essential across childhood to identify children’s strengths and difficulties and to provide adequate instruction and opportunities; assessment is a powerful tool to promote children’s development. This study aimed to investigate the hierarchal order of the Test of Gross Motor Development-Third Edition (TGMD-3) items regarding difficulty levels and the differential item functioning across gender and age group (3 to 5, 6 to 8, and 9 to 10 years old). Participants are 989 children (3 to 10.9 years; girls *n* = 491) who were assessed using TGMD-3. For locomotor skills, appropriate results reliability (*alpha* = 1.0), infit (*M* = 0.99; *SD* = 0.17), outfit (*M* = 1.18; *SD* = 0.64), and point-biserial correlations (*rpb* values from 0.14 to 0.58) were found; the trend was similar for ball skills: reliability (*alpha* = 1.0), infit (*M* = 0.99; SD = 0.13), outfit (*M* = 1.08; *SD* = 0.52); point-biserial correlations (*rpb* values from 0.06 to 0.59) were obtained. Two motor criteria: gallop, item-1, and one-hand forehand strike, item-4, were the most difficult items; in contrast, run, item-2, and two-hand catch, item-2, were the easiest items. Differential item functioning for age was observed in nine locomotor and ten ball skills items. These items were easier for older children compared to younger ones. The TGMD-3 has items with different difficulty levels capable of differential functioning across age groups.

## 1. Introduction

Across childhood, motor proficiency in fundamental motor skills (FMS) is considered the essential building block for children to learn more complex movements and participating in sports and physical activity. FMS requires large muscle groups to move the body from one place to another (e.g., run, jump, hop, leap, gallop, slide, skip) and receive or keep the objects under control (e.g., throw, catch, dribble, kick, strike) [1,2]. Additionally, these are crucial skills for children to be able to engage in play throughout childhood (i.e., tag, hopscotch, jumping rope, hoop throw, clap catch). Consequently, evaluating children’s motor proficiency is vital for health and education professionals.

The evaluation of children’s levels of motor proficiency requires that teachers and clinicians use valid and reliable tools for specific populations; the information obtained from the assessments ensures adequate care for children with delays, complementing the overall clinical evaluation, supporting the design and implementation of interventions, and the adaptation, when necessary, of the school curriculum. However, the assessment of motor delays across childhood is sometimes arduous, if the signs are subtle and complex to discriminate between typical and atypical development, since children present a broad repertoire of skills and could be less proficient in one specific area than in other (i.e., balance, fine motor skills, gross motor skills) [3,4,5].

Children develop at different rates and paths depending on their experience. Understanding these trajectories and outcomes is essential to designing the appropriate programs. Consequently, it is essential to use the appropriate instruments to identify children with typical and atypical development, detect changes over time, and predict health outcomes. Besides, the continuous investigation of different aspects of norm-referenced assessment validity can provide relevant information regarding the motor development status across different cultures [4].

One norm-referenced, globally recognized instrument is the Test of Gross Motor Development (TGMD-3) [6], a process-oriented assessment (i.e., scores are obtained from observing specific motor criteria established for each locomotor and ball skill). Children often experience the development of their locomotor and ball skills during daily activities and play throughout childhood. From the initial forms of gross motor play (i.e., travel from one place to another by imitating a horse, hopscotch, kicking a ball with a parent, throwing a ball into a box, jumping rope) to the more organized forms of small or large games (i.e., dodgeball, kick the can, capture the flag, tag, four square, freeze tag), these skills are part of the childhood. Therefore, FMS has ecological relevance to children’s daily lives [7], besides being related to the later acquisition of sport-specific motor skills [8,9].

Fundamental motor skill levels have been investigated for children worldwide using the various versions of the Test of Gross Motor Development [10]. The psychometrics of the TGMD-3 have been investigated in the United States [11,12], Brazil [13,14], Germany [15], Iran [16], Persia [17], Bosnia and Herzegovina [18], Spain [19], Italy [20], Canada [21], Greece [22] and Ireland [23]. The results provide relevant information about TGMD-3-appropriate indexes of reliability [11,12,13,14,15,17,18,19,24,25] and factorial structure [13,14,18,21]. For the current third edition, the psychometrics of the TGMD-3 items and the motor performance criteria (i.e., include preparatory movement, body position during the action, and follow-through) have been investigated only in the American sample regarding differential item functioning for gender and race, and satisfactory correlations (values above 0.25) were obtained, showing the validity of the item’s discriminative power [6].

To the best of the authors’ knowledge, no study investigated the TGMD-3 motor criteria regarding variance and hierarchal order. For example, two recent studies investigated the TGMD-3 skills discriminant power across gender and age [20,21]. However, the motor criteria were not addressed, and although the results provided insights into the skills difficulties, they did not provide information about item hierarchal difficulty. Gender plays a relevant role across childhood regarding motor proficiency; girls often demonstrate lower motor performance than boys [5,26,27,28]. Girls are less encouraged to engage in physical activity, negatively affecting their ball and locomotor skills [29,30]. To better understand the gender differences in motor proficiency, it is essential to investigate the motor skills criteria.

The TGMD-3 has 50 motor criteria, 3 to 5 for each skill; further understanding the items’ variance, hierarchal difficulty, and its differential function by gender and age will provide relevant information for practice. During instruction of a skill, a therapist or professor must know what the most difficult or easy criterion is and how difficult a specific criterion is regarding each skill for a child to master. Besides, some criteria could discriminate regarding different groups of children, such as girls and boys, or across ages, and determining what age to expect the child to acquire the more demanding proficient criterion of one skill is essential for intervention planning. Therefore, there is a need to examine the variation, hierarchal organization, and separation of the items that comprise the TGMD-3. This study aimed to investigate the hierarchal order of the TGMD-3 items regarding difficulty levels and the differential item functioning across gender and age groups (3 to 5, 6 to 8, and 9 to 10 years old).

## 2. Materials and Methods

### 2.1. Participants

This study included 989 children from 3 to 10.9 years old (mean (M) = 6.8, standard deviation (SD) = 2.1; girls: M = 6.9, SD = 2.0; boys: M = 6.9, SD = 2.1) from 22 schools (public and private), 10 cities in eight states (Amazonas, Pará, Ceará, Goiás, Mato Grosso do Sul, Minas Gerais, Santa Catarina, and the Rio Grande do Sul) from the five main regions of Brazil (North, Northeast, Central-West, Southeast, and South). Children with physical disabilities (i.e., vision impairment, hearing impairment, cognitive disability, physical disability, brain injury, spinal cord injury, psychological impairment) reported by parents, teachers, or caregivers, were excluded from the study. Participants’ descriptive statistics are presented in Table 1.

### 2.2. Instrument

We used the Test of Gross Motor Development-3rd Edition (TGMD-3) [6], validated for Brazilian children [13], to assess the locomotor skills (LOCS) and ball skills (BS) of children 3 to 10 years old. The LOCS subtest includes six skills (i.e., run, gallop, hop, skip, jump, and slide), and the BS subtest includes seven skills (i.e., one-hand strike, two-hand strike, one-hand dribble, two-hand catch, kick, overhand throw, and underhand throw). Each skill has 3 to 5 motor performance criteria describing the efficient movement pattern. The TGMD-3 was administered to each child following the standardized procedures that are recommended in the manual [6].

We used the Brazilian Economic Classification questionnaire [31] to verify the families’ SES. The questionnaire estimates families’ social classes by measuring family income, education levels, the number of durable consumer goods that family has (i.e., car, computer, TV, refrigerator), number of rooms in the house, and if they had a housekeeper working at home. These indicators categorize the SES into groups ranging from Category A (45 to 100 points)—most favored, to Category D-E (from 0 to 16 points)—least favored low-income families.

### 2.3. Procedures

The sample size was estimated considering a minimum of 10 subjects per threshold [32,33]; in the present study, each item has two thresholds (e.g., three response categories—1 = 2), and the locomotor and ball skills dimensions have 23 and 27 items, respectively. A minimum of 598 participants for locomotion and 702 for ball skills were estimated considering participants’ waiver data loss of 30%. Additionally, we also extended the number of participants assessed, aiming to strengthen the sample representativity of different geographic regions, considering that Brazil is a continental country.

For data collection, parents or legal guardians of the child were informed, by letter, of the study’s aims and procedures. The informed consent was sent home to the parents interested in participating; children who returned the signed informed consent were assessed after being asked and verbally agreeing to participate in the study. Parents/legal guardians completed a demographic information questionnaire. The TGMD-3 was administered at school by three members of the research team. The assessment was conducted with two children simultaneously, following the protocol procedures regarding instruction, demonstration, practice trial, and formal trials. The assessment of all children was video recorded for later motor performance scoring. Two trained raters, with extensive coding experience using the TGMD-3, coded all video records (N = 989). Considering a minimum acceptable reliability (ICC) (ρ0 = 0.6), expected reliability (ICC) (ρ1 = 0.75), significance level (α) = 0.05, power (1 − β) = 80%, and number of repetitions per subject (k) = 2, the intra-rater sample size calculated was 102 [34]. Then, 102 children were randomly selected for intra-rater and inter-rater reliability analysis; intra-rater (with two-month interval; ICC = from 0.70 to 0.90) and inter-rater (ICC = 0.85 to 0.99) reliability were high. The age band for the statistical analyses was organized into three groups, considering school organization in Brazil, pre-school (3 to 5 years old), first to third grades—the first cycle of fundamental education I (6 to 8 years old), and fourth and fifth grades—the second cycle of fundamental education I (9 to 10 years old).

### 2.4. Data Analysis

The sample size was calculated using Epilnfo statistical software (version 70), considering an expected 50% frequency, a design effect of 1.5, a 95% confidence level, a 4% acceptable margin of error, and 10 to 15% possible losses. The number of subjects comprising the final sample was between 990 and 1.035 children.

Data entry errors and inconsistent records were identified and corrected. Data cleaning was performed using Excel—360 and SPSS, version 22.0 [35]. The LOCS and BS were examined using Rasch analysis; in this model, the participant’s response to a given item is a function of participant skill and item difficulty. The analysis provides the participants’ response to an item location, latent traits, and indexes of items’ fit to the measurement model [36]. The scores of each child in the two formal trials of the TGMD-3 were analyzed; therefore, the data responses for each item (i.e., motor performance criteria) range from 0 to 2 (0 = the child was not successful in the two trials; 1 = the child was successful in one trial; 2 = the child was successful in two trials). Since the response possibilities to the TGMD-3 items range from 0 to 2 points, the extension of the Rasch model to polytomous items, Andrich’s [37] graduated scale for the model, was used.

The fit of the items to the model was examined using the standardized indices of infit mean-square, measuring the discrepancy between the model’s prediction and the observation of the participants’ responses by weighting with item information. Outfit mean-square is not weighted by item information, thus proving more sensitive to extreme residuals (outlier cases), where the item misfit or discrepancy occurs far from the level of the subject’s latent trait [34,38]. Items with infit mean-square and outfit mean-square values close to 1.00 are the ones that most contribute to the construction of the measure. Values below 0.50 are less productive, but do not degrade the measure. Values between 1.50 and 2.00 do not contribute much, but also do not degrade the quality of the measure. Values above 2 represent noise, or item variance not explained by the factor effect [34]. Therefore, values between 0.50 to 1.50 for infit and outfit were considered adequate [34] and were adopted in the study. Additionally, the point-biserial correlation discrimination estimates (ρpbde) were used. In Rasch analysis, the ρpbde is a helpful diagnostic indicator of data miscoding or item miskeying. Several recommended interpretations were adopted. First, negative or 0 values indicate items or persons with response strings that contradict the variable [34]. Second, positive values are less informative than infit and outfit statistics [34]. Third, if negative or low values were observed, this was an indication of items that were too easy or too difficult, and they were performed poorly by students with higher levels of ability [39]. Fourth, the items must have point-biserial values ranging from 0.30 to 0.70 [40].

Differential item functioning (DIF) was used to identify items that function differently concerning specific subgroups within the population of interest. An item exhibits DIF when participants who possess the same ability level, but come from different subgroups, are not equally likely to answer it correctly or endorse a particular response category [41]. The DIF was accessed for gender and age groups, contrasting the difficulty parameter between the groups. The difference in the difficulty parameter (DIF contrast) between groups larger than a logit of 0.64 was considered as evidence for the occurrence of differential item functioning, with an acceptable effect size [42]. Winsteps 3.709 software [42] and Software R [43] were used to conduct the analyses, in addition to the functions implemented by the psych R [44] and lavaan R [45] packages.

## 3. Results

Hierarchic order of the TGMD-3 items: item adjustment and motor performance difficulty.

### 3.1. Locomotor

Participants’ scores on the LOCS, using the Rasch measure, ranged from −4.06 to 2.96 (M = 0.l8; SD = 0.82), with a mean standard error of 0.33 (SD = 0.07). The participants’ mean infit was 0.98 (SD = 0.44), and the mean outfit was 1.18 (SD = 1.27). The person separation coefficient was 2.09; values above 2 are considered adequate [46], and the reliability of the estimates of people’s abilities was 0.81, indicating that participants were separated into groups of two levels of motor performance, with high reliability. The reliability of the scale was 1.00, with the separation index of 17.86, the mean infit of the items was 0.99 (SD = 0.17), and the mean outfit was 1.18 (SD = 0.64), with a standard error mean of 0.05 (SD = 0.07).

The item-person map represents children’s LOCS performance on the TGMD-3, with the difficulty level of the items listed from least to most difficult, are presented in Figure 1. This map is based on the calibration of the items and illustrates performance on a continuum. In general, children showed a latent trait level near the item difficulty (theta mean = 0.18; difficulty mean = 0.0). Further, it was observed that most items were grouped between +1 and −1.3 logit, approximately, so the model showed a high precision in the estimate of participant’s theta located in these limits. The test curve supports this information (Appendix A).

Regarding the analyses of each LOCS item, Table 2 presents the results of the difficulty, infit, outfit, and point biserial correlations. It is possible to observe that the hardest items were gallop, item 1 (d = 1.50), hop, item 3 (d = 1.35), and horizontal jump, item 2 (d = 1.21). The easiest items were run, item 2 (d = −2.45), run, item 3 (d = −1.27), and hop, item 4 (d = −1.05). Three items had adjustment problems detected by the outfit (run, items 2, 3, and 4); such items also had low correlations with the latent trait under investigation, showing a larger pattern of unexpected performance. The point-biserial correlations of the items ranged from 0.14 to 0.58 (M = 0.45; SD = 0.11), with 95.7% of the correlations above 0.30.

Due to adjustment problems in run items 2, 3, and 4, the model was reanalyzed, and the indexes did not change substantially regarding participants’ scores (mean infit = 0.99, SD = 0.42; mean outfit = 1.17, SD = 0.78; person separation coefficient = 2.08; reliability = 0.81) and items (mean infit = 0.98, SD = 0.16; mean outfit = 1.07, SD = 0.34; person separation coefficient = 16.76; reliability = 1.00).

### 3.2. Ball Skills

Participants’ scores on the BS, using the Rasch measure, ranged from −4.15 to 3.93, with a mean of −0.01 (SD= 0.69) and a mean of the standard error of 0.28 (SD =0.05). The infit mean was 0.99 (SD = 0.13), and the outfit mean was 1.08 (SD = 0.70). The person separation coefficient was 2.12. The reliability of the estimates of abilities was 0.82. The reliability of the scale was 1.00, with the separation index of 16.74, the mean infit of the items was 0.99 (SD = 0.13), and the outfit was 1.08 (SD = 0.52), with a standard error mean of 0.05 (SD = 0.07).

The item-person map of the children’s BS performance is presented in Figure 2. Most children showed a latent trait level similar to the item difficulty (theta mean = −0.01; difficulty mean = 0.0). Further, it was observed that most items were approximately grouped between +1 and −1 logit. Therefore, the model accurately estimated the participant’s theta located in these limits. The test curve information (Appendix A) reinforces this evidence.

Concerning the analyses of each BS item, Table 3 presents the difficulty, infit, outfit, and point-biserial correlation results. The analyses showed that the hardest items were strike with one hand, item 4 (δ = 1.13), underhand throw, item 2 (δ = 0.9), and dribble, item 1 (δ = 1.21). On the other hand, the easiest items were catch, item 2 (δ = −1.44), kick, item 4 (δ = −1.27), and catch, item 1 (δ = −1.05). One item (two-hand catch, item 1) showed outfit misfit (3.48), indicating an unexpected performance pattern and a small correlation with the BS factor. Items’ point-biserial correlations ranged from 0.06 to 0.59 (M = 0.42, SD = 0.13), with 96.3% of the items having factor correlations above 0.30.

Considering the outfit problems presented in two-hand catch, item 1, the model was reanalyzed, and the indexes did not change markedly. (The new indexes for participants’ scores were: mean infit = 1.00, SD = 0.28; mean outfit = 1.02, SD = 0.50; person separation coefficient = 2.15; reliability = 0.82. The new results for items were: mean infit = 1.00, SD = 0.12; mean outfit = 1.02, SD = 0.24; person separation coefficient = 16.62; reliability = 1.00.)

### 3.3. Differential Item Functioning: Gender and Age Groups

Table 4 and Table 5 present the DIF of the LOCS and BS items, respectively, by gender and age groups. No DIF was found for LOCS and BS by gender (logits values below 0.64). The response pattern was similar for girls and boys.

Regarding LOCS by age group, eight items had differential functioning, run (items 1, 2, and 3), hop (item 4), skip (item 2), jump (item 4), and slide (item 3 and 4). All these items were more difficult for younger children (3 to 5 years old).

Regarding BS by age group, ten items had differential functioning, two-hand strike (item 2), two-hand catch (item 2), two-hand catch (item 3), one-hand dribble (item 2), one-hand dribble (item 3), kick (items 1 and 4), overhand throw (item 4), underhand-throw (item 2), and underhand-throw (item 4). All these items were more difficult for younger children (3 to 5 years old).

## 4. Discussion

This study aimed to investigate the hierarchic order of the TGMD-3 items regarding difficulty levels and the differential item functioning across gender and age group (3 to 5, 6 to 8, and 9 to 10 years old). Although recent studies investigated the TGMD-3 skills item difficulty [6,21], all of these considered the motor skills as items, and not the criteria that comprised the motor skills. To the best of the authors’ knowledge, no study investigated the TGMD-3 motor criteria regarding hierarchal difficulty across gender and age.

In this study, Rasch rating scale model analyses showed adequate indexes for participants’ scores. The person separation coefficient for LOCS and BS dimensions were acceptable (values of 2.09 and 2.12, respectively); values above 2 are considered adequate [47]. Regarding the reliability of the estimates of people’s abilities, the results also showed adequate indexes for LOCS and BS; participants were separated into groups of two levels of motor performance, with high reliability [46]. These results indicated that the estimated trend could be adequately replicable.

Regarding the items, Rasch analyses also showed adequate indexes for the person separation coefficient and the scale’s reliability in both LOCS and BS [46]. Furthermore, analyses showed adequate infit for all items [34]. Infit is a metric that assesses adequacy of the response patterns of the individuals that have the same level of a latent trait to the item difficulty (i.e., if the pattern responses are adequate to level of the latent trait and the item difficulty). It means that, in the present study, the response patterns of children showed an adequate adjustment between the trait latent level and the difficulty of the respective item. Considering the perspective of test precision, it is desired the infits of an instrument be adequate [34].

The analyses showed items with inadequate outfit. Items 3 and 4 for run, and item 1 from two-hand catch, showed higher unexpected performance than other items (outfit values = 3.82, 1.79, and 3.48 respectively). This means that the items are less robust to extreme cases and, for example, children with high level of the latent traits for locomotor and ball skills performed unsatisfactorily for these items, and vice versa. According to Linacre [41], outfit problems are less of a threat to measurement and can be fixed. The recommended guidelines for fix this problem are investigating the problematic item or person, removing the item, retesting the new models, observing if the exclusion of the item or person causes further distortion or denigrates the measure, and deciding if the measure is good enough [41]. In this study, we removed the misfit items and ran a new analysis; the further analysis did not improve the indexes.

We also observed low point biserial correlation values in run, item 3 (ρ_pbde_ = 0.14), and two-hand catch, item 1 (ρ_pbde_ = 0.10). The point biserial correlation results, specifically, regarding run, item 3, and two-hand catch, item 1, indicated a very weak relationship with their respective locomotor and ball skills latent trait. This means that the item can fail to discriminate those individuals with distinct latent traits [38].

Considering the adequate infit and outfit for most of the items observed in the present study, from this point forward, we focused on discussing the items with higher and lower difficulty, in both LOCS and BS dimensions. Regarding the LOCS items, the gallop, item 1, showed a higher difficulty level, followed by hop, item 3, horizontal jump, item 2, and skip, item 2. All these items are related to arm action to help in the movement. These items require that children utilize the arms to produce force and/or maintain the body balance, aiming to help in projecting the body forward. These aspects seem to make these items more difficult. Supporting this evidence, the results reported by Ulrich [6] showed that these were the items in which the children demonstrated a lower percentage of mastery, compared to all other items.

On other hand, run, items 2 and 3, and hop, item 4, were those that presented lower difficulty. It is plausible to consider that the lower difficulty item in the TGMD-3 was the flight phase in the running skill, because it reflects an inherent part of the movement, without which it would not be possible to perform the skill (i.e., the flight or float phase is a fundamental characteristic that distinguishes a walk from a run) [47,48,49]. In the validity study, Ulrich [6] also observed that a high percentage of children achieved mastery of this item (from 94% at 3 years old to 100% at 10 years old). Regarding run, item 3, it is plausible to consider that the absorption or landing phase is also a natural consequence of the running skill, requiring that the foot be placed on the surface, usually with the heel or toes, to absorb impact forces and avoid injuries [50,51,52]. The flat-footed stance in the running skill can frequently be observed in younger children (i.e., around 2 to 3 years), when the children are having their initial experiences with running [53,54,55]. Therefore, these run items seem not to be pertinent for the age band targeted by the TGMD-3. However, from a conceptual point of view, the run is a fundamental motor skill in childhood; is present in the most of the oriented tasks or free play activities, both at in and outside school [56,57,58]. Our results suggest that these run items, 2 and 3, are not criteria to consider when we are focusing on distinguishing the children with high or low motor ability, but we understand that these items must be retained in the test.

Regarding hop, item 4, only the capability of the children for performing hops with preferred the leg, for four times consecutively, is assessed, focusing on the product of the movement. Neither qualitative aspect of the movement is required; therefore, completing this item is less difficult. Furthermore, in Brazil, children frequently are enrolled in free play activities that require hops for consecutive times, for example, amarelinha (hopscotch), pula corda (jumping rope), or elástico/trancelim (jumping an elastic rope) [57,59,60]. The prior experience of the children regarding these motor tasks can explain these results.

Concerning the ball skills, the one-hand forehand strike, item 4, was the most difficult item. This item requires a follow-through movement across the body (i.e., contralateral movement of follow-through of the arm), which may constrain the amplitude of movement if less force is applied at the beginning of movement. Although other TGDM-3 ball skill items include a contralateral follow-through movement (e.g., overhand throw, item 4), these do not require holding or retaining an implement. Holding a paddle while performing the movement adds an additional demand that can make this TGMD-3 item more difficult for children. In addition, the lack of practice may explain this result. In general, activities that involve striking a ball with a paddle occur less frequently, or are not included in the physical education curriculum in most Brazilian public schools [61,62]. Thus, this may explain the high difficulty of this item for the children assessed in this study.

The second most difficulty ball skill item was the underhand throw, item 2. The obtained result is plausible, considering the numerous demands required to complete the task. This item requires a coordination of the upper and contralateral low limb, with a body projection forward. Besides, the movement of the throwing hand is focused on hitting the ball into a prior specified target (i.e., in the wall), which can increase the difficulty of the task. In TGMD-3, the overhand throw, item 3, also requires coordination between the upper and contralateral lower limb and body projection forward, but different from the underhand throw, item 2, it does not require throwing a ball into a wall.

Another high-difficulty item was the dribble, item 1; this item requires that children contact the ball with one hand at waist level, demanding visuo-motor coordination with the ball in motion. In this situation, the children must combine sensory information with motor response, coordinating the eye and the hand movements, trying to achieve temporal and spatial accuracy for manually intercepting the ball in motion [63]. If the children fail to combine these actions, the performance tends to be poor. In addition, Brazilian children have showed a low practice frequency in sports such as handball and basketball [64], which require contact and ball manipulation. These factors can explain the observed results regarding the difficulty of this item for Brazilian children.

The results showed that the less difficult items were two-handed catch, items 1 and 2, and kick, item 4. These two items cited from the catch skill seem to require a low motor demand; the first, with the arms extending to reach the ball, and the other, a ready or waiting position, in this case, to remain standing in front to the ball. This can explain the low difficulty of both items. Regarding the kick, item 4, kicking a ball with the instep or inside of the preferred foot is a not so difficult task for Brazilian children, considering their further experiences throughout childhood with soccer and futsal practices at school and during outdoor free play [64]. The children’s experiences in these sports can explain the results concerning the low difficulty of this item. This argument is reinforced when we compare our results with those reported by Ulrich [6]; in general, a small group of American children, from 3 to 6 years old, demonstrated mastery in kicking a ball with the instep or inside of the preferred foot (percentages varying from 0% to 23%). Older American children (7 to 10 years old) also demonstrated a lower percentage of mastery, especially among girls (from to 22% to 48%) [6].

### 4.1. Differential Item Functioning between Gender and among Age Groups

No one item showed differential item functioning between gender; the response pattern for both locomotor and ball skills items was similar in girls and boys. It is important to emphasize that although the literature addresses the differences in performance between girls and boys [1,5,28], at this moment, we are not disagreeing with what the literature points out. However, the differential item functioning analysis points out that the pattern of responses of boys and girls related to the difficulty of the item is similar, and not the average of motor performance. For example, the way girls and boys respond to throwing skill development is similar.

Concerning locomotor item functioning across age groups, the analyses showed that, for example, some run, hop, slide, skip, and jump items were easier for older children compared to the youngest ones (3 to 5 years old). Consequently, with an increase in age, the percentage of children demonstrating mastery in these items also increased; similar evidence was reported by Ulrich [6]. For example, in the present study, 22% of the 3-year-old and 55% of the 4-years-old children demonstrated mastery in hop, item 4, whereas 89%, 97%, and 97% of the children at 8, 9, and 10 years old, respectively, showed mastery. Further, 15% of the children reached mastery in slide, item 4, at 3 years old, whereas 79% to 92% demonstrated mastery at 6 to 10 years old.

Ball skills items also showed differential functioning. Two-hand catch, item 2, and kick, item 4, were even easier for the older children groups compared to the younger children. Furthermore, two-hand strike, item 2, two-hand catch, item 3, dribble, items 2 and 3, kick, item 1, and overhand throw, item 4, were easier for older children. The results from Ulrich [6] also confirmed the changes in the percentage of children that demonstrated mastery according to age, especially regarding the two-hand catch, item 3, dribble, item 2 and 3, and overhand throw, item 4. For example, no one children at 3 years old and only 7% those at 4 years old and demonstrated mastery in dribble, item 3, while 75% to 89% of children at 7 to 10 years old reached mastery. In the same way, 2% of children at 3 years old and 11% at 4 years old demonstrated mastery, while 72% to 89% of children at 8 to 10 years old reached it. This evidence gives support to the developmental characteristic of these TGMD-3 items.

### 4.2. Implications

The TGMD-3 has items with hierarchical levels of difficulties according to age, allowing the identification of specific challenges within each skill. Teachers and therapists will benefit from this information by understanding the motor criteria within each skill that require more instruction and practice, or specific strategies for children, especially the younger ones, to achieve skill mastery. Understanding motor criteria hierarchy within each skill across different ages may also help teachers and therapists to design more effective intervention programs to enhance the motor development of children with delays. Another possible implication of the present study is regarding TGMD-3 research recommendations. The nondifferentiation in function for gender and the observed differentiation across different ages emphasized the importance of addressing the motor criteria in further studies across gender and sex to build additional knowledge into the paths of children’s developmental trajectories in different cultures.

### 4.3. Strength

In advancing the previous study, the analysis of the results from the Rasch method provided important evidence for TGMD-3′s hierarchy, difficulty, and item differential functioning. Furthermore, the separation into distinct performance groups showed the ability of TGMD-3 to detect different levels of motor performance in all age groups. The combination of information acquired in this study shows the sensitivity of TGMD-3 in detecting changes, collecting crucial information for identifying children who need further support, and referring children to a specific clinical intervention.

### 4.4. Limitations

The evidence highlighted in this study must also be observed from the perspective of its limitations. The first limitation is related to the sample enrolled. It was composed of typically developing children. Another limitation is the lack of previous studies examining item psychometrics, which restrains our capacity for comparison with other samples.

### 4.5. Future Directions

Item difficult varies according to the children’s latent trait levels. For example, a given item can be easy for typical children, but, more difficult for atypical children (i.e., children with neuromotor disabilities). It is plausive, therefore, to test the hierarchy, difficulty, and item differential functioning among typical and atypical children. Including children with disabilities could provide different trends in the item’s difficulty, broadening the knowledge about where these children are localized in the locomotor and ball skills latent trait continuum. In addition, the models showed a high precision for estimating the participant’s theta located within a reasonably wide limit of the theta continuum. However, it is important to investigate how the inclusion of items with more difficulty can increase the precision for theta estimating those children with high or very high motor performance.

## 5. Conclusions

The results observed in the present study showed that the TGMD-3 contains locomotion and ball skills items with different levels of difficulty; therefore, it has the ability to identify motor changes in Brazilian children, with different levels of performance in research and educational contexts. Furthermore, we verified that the TGMD-3 showed differential item functioning in the performance of distinct age groups for children, indicating that the instrument is suitable for monitoring motor development over time during childhood. These findings offer reliable support to health and education professionals dedicated to carrying out scientific research aimed at assessing and monitoring children’s motor behavior, as well as planning interventions appropriate to the level of child development according to age.

## Figures and Tables

**Figure 1 ijerph-19-08667-f001:**
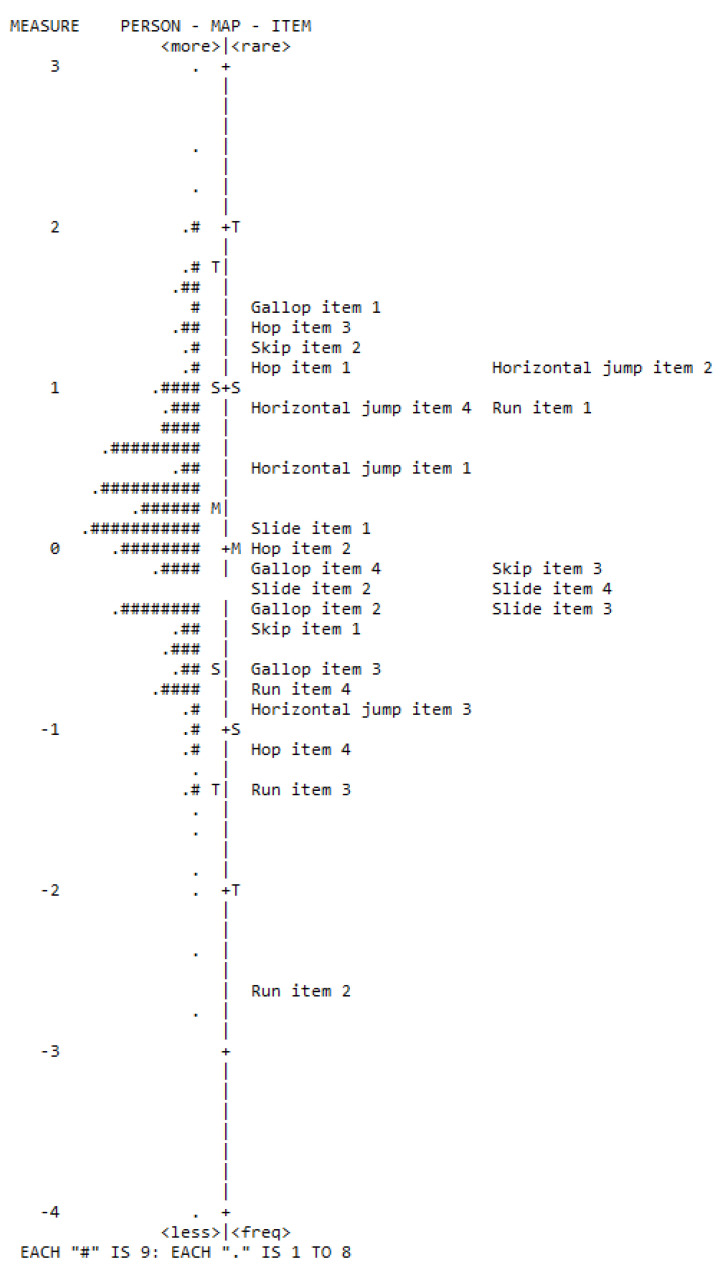
Person-item map: score analysis of the 23 TGMD-3 locomotor skills items. Note: M = mean of person or item distribution; S = one standard deviation from the person or item mean; T = two standard deviations from the person or item mean; # = several persons or items.

**Figure 2 ijerph-19-08667-f002:**
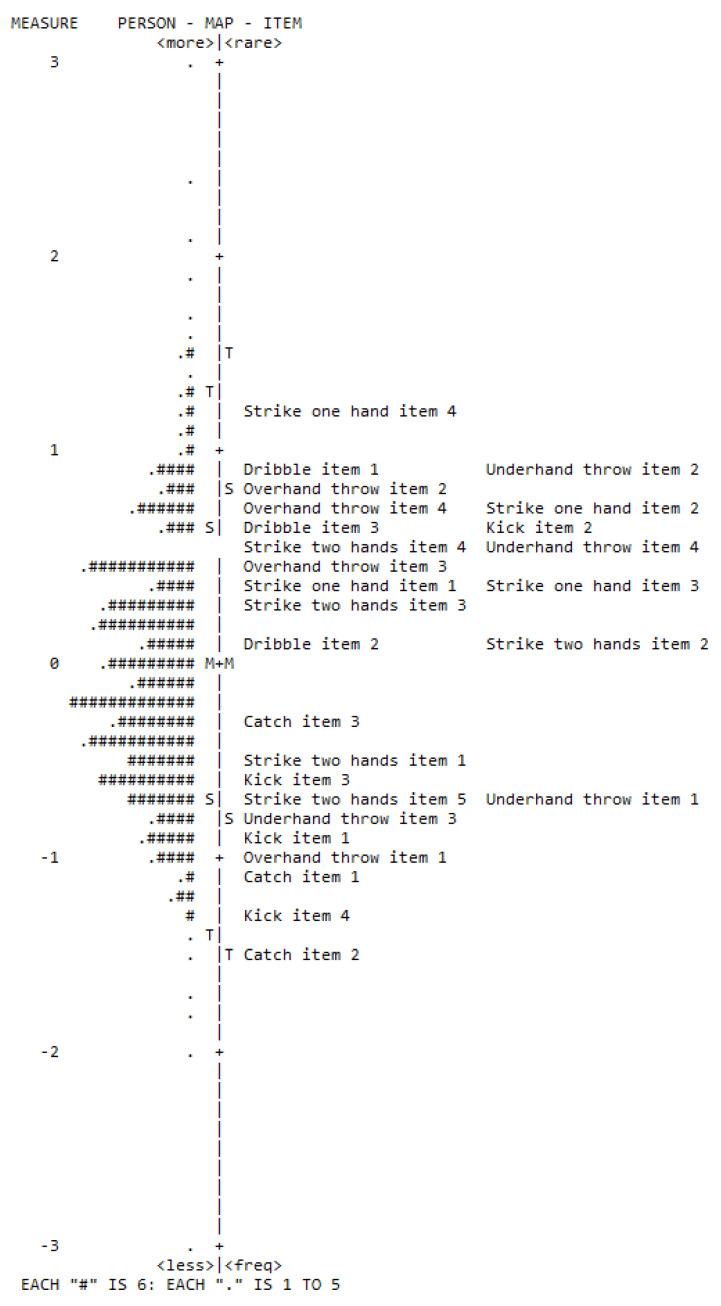
Person-item map: score analysis of the 27 TGMD-3 ball skills items. Note: M = mean of person or item distribution; S = one standard deviation from the person or item mean; T = two standard deviations from the person or item mean; # = several persons or items.

**Table 1 ijerph-19-08667-t001:** Descriptive statistics of participants.

Characteristics of Participants	N = 989 *n* (%)
Gender	
Boys	498 (50.65)
Girls	491 (49.64)
Age (in years)	
3	37 (3.74)
4	115 (11.62)
5	167 (16.88)
6	131 (13.25)
7	128 (12.95)
8	128 (12.95)
9	158 (15.97)
10	125 (12.64)
Schools	
Private schools	80 (8.09)
Public schools	909 (91.91)
Children’s educational level	
Pre-school	310 (31.34)
Fundamental school	679 (68.66)
Socioeconomic status	
High	98 (9.90)
Middle	296 (29.92)
Middle-low and poor	595 (60.18)

Note: *n* = number.

**Table 2 ijerph-19-08667-t002:** Percentage of response, Item Difficulty, Infit and Outfit Indexes, and Point-Biserial Correlations with the Locomotor Skills Scale Factor.

Skills	Item	Response Categories (%)			Difficulty	Infit	Outfit	Point-Biserial
		0	1	2				
Run	1	63.3	12.1	24.6	0.83	0.96	1.08	0.47
	2	2.0	1.4	96.6	−2.45	1.04	1.51 *	0.29
	3	10.3	2.9	86.8	−1.27	1.45	3.82 *	0.14
	4	17.1	11.2	71.7	−0.78	1.24	1.79 *	0.29
Gallop	1	81.7	6.8	11.5	1.50	0.87	0.88	0.46
	2	29.1	13.4	57.4	−0.27	1.19	1.48	0.35
	3	19.8	10.2	70.0	−0.67	1.01	1.08	0.46
	4	31.0	14.0	55.0	−0.20	1.05	1.24	0.44
Hop	1	75.6	4.1	20.2	1.09	0.76	0.68	0.56
	2	38.6	12.9	48.4	0.04	1.00	0.98	0.49
	3	80.6	5.0	14.5	1.35	0.75	0.64	0.54
	4	12.7	7.7	79.6	−1.05	0.99	1.37	0.42
Skip	1	29.0	6.8	64.2	−0.35	0.88	0.83	0.55
	2	78.9	2.8	18.3	1.19	0.80	0.68	0.54
	3	35.0	7.3	57.7	−0.15	0.98	1.14	0.49
Jump	1	49.5	16.6	33.9	0.44	0.88	0.82	0.56
	2	67.0	18.8	14.2	1.21	1.08	1.06	0.38
	3	12.7	15.6	71.7	−0.96	1.19	1.47	0.32
	4	67.0	9.2	23.8	0.89	0.87	0.84	0.53
Slide	1	39.2	20.6	40.1	0.17	1.14	1.34	0.38
	2	34.1	15.4	50.6	−0.08	0.90	0.90	0.55
	3	27.4	14.2	58.4	−0.32	0.84	0.78	0.58
	4	31.7	15.5	52.8	−0.15	0.83	0.80	0.58

Note: * unadjusted item by outfit.

**Table 3 ijerph-19-08667-t003:** Percentage of Response, Item Difficulty, Infit and Outfit Indexes, Point-Biserial Correlations with the Ball Skills Scale Factor.

Skills	Item	Response Categories (%)			Difficulty	Infit	Outfit	Point-Biserial
		0	1	2				
Two-handstrike	1	30.2	8.0	61.8	−0.49	1.13	1.19	0.35
	2	47.6	13.7	38.7	0.11	1.16	1.24	0.35
	3	51.3	16.4	32.4	0.27	0.89	0.87	0.54
	4	62.7	14.5	22.9	0.63	0.93	0.94	0.50
	5	11.3	35.3	53.4	−0.95	1.07	1.13	0.30
One-handstrike	1	57.3	10.3	32.4	0.35	0.96	0.89	0.50
	2	66.7	10.2	23.1	0.67	0.79	0.73	0.59
	3	54.7	19.1	26.2	0.45	0.81	0.77	0.59
	4	81.4	4.9	13.8	1.13	0.90	0.70	0.49
One-handdribble	1	71.1	10.9	18.0	0.87	1.02	0.97	0.43
	2	50.4	6.4	43.3	0.07	0.89	0.84	0.55
	3	62.7	14.9	22.4	0.64	0.85	0.76	0.56
Two-handCatch	1	15.9	5.2	79.0	−1.05	1.26	3.48 *	0.06
	2	8.5	8.6	82.9	−1.44	0.95	0.85	0.37
	3	31.6	20.8	47.5	−0.28	0.84	0.81	0.57
Kick	1	19.3	11.2	69.5	−0.84	1.16	1.37	0.27
	2	64.1	10.8	25.1	0.59	0.84	0.76	0.57
	3	21.4	18.5	60.1	−0.67	1.03	0.98	0.41
	4	10.9	8.6	80.5	−1.27	0.91	0.76	0.43
Overhandthrow	1	19.8	5.6	74.6	−0.88	1.06	1.31	0.34
	2	72.3	7.4	20.3	0.81	0.91	0.77	0.52
	3	62.6	11.3	26.1	0.55	0.97	0.94	0.49
	4	66.4	10.7	22.9	0.67	1.01	1.03	0.44
Underhandthrow	1	24.7	11.0	64.3	−0.64	1.09	1.19	0.35
	2	71.7	11.0	17.3	0.90	1.12	1.39	0.33
	3	17.3	18.9	63.8	−0.84	1.20	1.43	0.22
	4	64.6	12.7	22.6	0.66	1.09	1.17	0.38

Note: * unadjusted item by outfit.

**Table 4 ijerph-19-08667-t004:** Differential Item Functioning (DIF) of locomotor items according to gender and age group.

Skills	Item	Gender	Age Groups Comparison (Logit Scores)		
		Boys vs. Girls	3–5 yrs vs. 6–8 yrs	3–5 yrs vs. 9–10 yrs	6–8 yrs vs. 9–10 yrs
Run	1	0.27	−0.53	−0.77 *	−0.24
	2	−0.52	1.86 *	0.60	−1.26 *
	3	−0.20	0.67 *	0.62	−0.05
	4	0.20	−0.40	−0.56	−0.17
Gallop	1	0.00	−0.44	−0.25	0.19
	2	0.00	−0.35	−0.34	0.01
	3	0.00	−0.14	−0.17	−0.03
	4	−0.09	−0.03	0.07	0.11
Hop	1	0.08	−0.17	−0.26	−0.08
	2	−0.07	0.22	−0.15	−0.36
	3	0.00	−0.30	−0.34	−0.03
	4	−0.14	1.10 *	1.46 *	0.38
Skip	1	−0.19	0.10	0.38	0.29
	2	0.08	0.97 *	0.96 *	0.00
	3	−0.23	−0.07	0.16	0.26
Jump	1	0.20	0.12	0.06	−0.08
	2	−0.10	−0.43	−0.49	−0.06
	3	−0.11	−0.51	−0.53	−0.03
	4	0.00	−0.48	−0.56 *	−0.08
Slide	1	0.11	−0.18	−0.13	0.04
	2	0.14	0.40	0.39	−0.02
	3	−0.05	0.35	0.70 *	0.33
	4	0.05	0.44	0.74 *	0.28

Note: * DIF according to age group.

**Table 5 ijerph-19-08667-t005:** Differential Item Functioning (DIF) of the Ball Skills by gender and age group.

Skills	Item	Gender	Age Groups Comparison (Logit Scores)		
		Boys vs. Girls	3–5 yrs vs. 6–8 yrs	3–5 yrs vs. 9–10 yrs	6–8 yrs vs. 9–10 yrs
Two-hand strike	1	−0.18	−0.07	−0.09	−0.02
	2	−0.16	−0.59	−0.87 *	−0.28
	3	0.22	0.33	−0.10	−0.43
	4	0.07	0.02	−0.24	−0.26
	5	−0.09	0.00	−0.08	−0.08
One-hand strike	1	0.00	0.30	0.56	0.25
	2	−0.05	0.50	0.48	−0.01
	3	0.38	0.31	0.71	0.40
	4	0.14	−0.40	−0.50	−0.10
Two-hand catch	1	−0.38	−0.28	−0.60	−0.31
	2	−0.03	0.33	0.78 *	0.46
	3	−0.18	0.97 *	1.41 *	0.43
One-hand dribble	1	−0.23	−0.38	−0.36	0.01
	2	−0.05	0.81 *	1.44 *	0.63
	3	−0.14	0.79 *	1.19 *	0.40
Kick	1	−0.09	−0.37	−1.04 *	−0.67 *
	2	0.22	−0.09	−0.06	0.03
	3	−0.09	−0.22	−0.36	−0.14
	4	0.10	0.50	0.95 *	0.45
Overhand throw	1	−0.20	0.21	0.20	−0.01
	2	0.26	0.00	−0.21	−0.21
	3	0.23	−0.25	−0.07	0.18
	4	0.17	−0.53	−0.79 *	−0.26
Underhand throw	1	−0.09	−0.03	0.09	0.12
	2	0.20	−0.81 *	−0.63	0.19
	3	−0.09	0.07	0.29	0.22
	4	0.00	−0.79 *	−0.78 *	0.01

Note: * DIF according to age group.

## Data Availability

The original contributions presented in the study are included in the article/Appendix A; further inquiries can be directed to the corresponding author/s.

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
