# Peer review of "Test of Gross Motor Development-3: Item Difficulty and Item Differential Functioning by Gender and Age with Rasch Analysis"

_ijerph, 2022, doi:10.3390/ijerph19148667_

Round 1

Reviewer 1 Report

Introduction

Overall the introduction provides a strong rationale for item analysis however, I think the introduction needs to shift from a global perspective to the gap ion the literature regarding reasons of sex and age gaps.

Line 32 – you discuss motor proficiency then go straight in motor delays across multiple domains (Balance, fine motor, gross motor). Given the manuscript only addresses Fundamental Motor Skills different terminology and definitions is warranted (refer to Logan et al. 2018)

Line 47: do -> to

Lines 53-55- The assessment has also been examined in countries such as Greece, Portugal, Canada etc. a statement of the worldwide examination of children’s gross motor skills is well documented would suffice here.

Lines 66-77: I would expand on these statements in your introduction. Specially, transform the introduction away from the global relevance of the TGMD-3 and more about the sex differences and age differences that are prevalent in the literature. Then given the results of this study discuss why the sex differences were not existent in this sample, particularly for ball skills. Also, you mention the practical implications for therapists and teachers here in the introduction, but do not follow up or expand in the discussion for a potential implications section.

Lines 80-83- move to data analysis or methods section

Lines 89-92 – repetitive from procedures in methods. Remove and keep in procedures.

In Procedures authors need to discuss the age band splits for analysis.

Line 117- Why were only 100 children selected for the ICC. Typical reliability calls for 20-30% of a sample and this seems low. Were raters trained against a gold standard?

Results

Line 129- trial instead of trail

Figures 1 and 2 – the outputs of the person item map need to be in English and clear to read.

 Table 4. – Why is there no data after hop for 6-8 yr olds and 9 to 10 yr olds?

Discussion

Lines 228-231 ;Lines 241- 246 – these are very good points from the results of the current study however, expanding upon these examining implications and address the gaps in intervention and physical education would be additive. Authors should consider and “implications” section.

Line 265-TGMD-3 is misspelled

Line 268; Line 277, Line 285 – citations needed

Lines 298-299 – the fact there was no differences for sex is a very important finding I feel the authors just “glossed” over. I feel this should be a large part of the discussion. I am however concerned that in the Rash Modeling you did not examine the data from a sexX age interaction for item analysis (example 3-5 year old girls/ 3-5 year old boys) and run the data using that type of clustering. Was this done or have authors considered this point? I am concerned the data of no sex differences needs further examination and would be very additive to practical application literature.

Line 305- percentage

Line 328-331: If authors add the much needed implications section expanding on this state in regard to skill difficult and potential skill progressions and skill criteria to focus on in intervention based on the results of the item analysis is needed.

Line 347- list skills

Author Response

Dear Reviewer,

We thank you for allowing us to review our manuscript. The reviewers' observations were considered and the responses are in attachment. Corrections can be seen in blue in the manuscript. All suggestions have improved the quality of the manuscript and we are grateful for that.

Reviewer 2 Report

Thank you for the opportunity to review this interesting manuscript. The study wants to investigate the hierarchic order of the TGMD-3 items. It would like to determine the order based on the difficulty levels and differential item functioning across gender and age groups. This is an interesting study especially because some clarification are created for an important test and some indication are provided increasing the quality of future works. Consequently, I suggest the publication of this manuscript after major revisions.

Major revisions are especially for the introduction. It is hard to read and to arrive to the objective of the study because the speech is not linear.

Abstract

Line 13: please provide what TGMD-3 is.

Line 13: please, avoid to use sex, better if the term gender is adopted. Please, double check it through the manuscript.

Introduction

Line 28: perhaps “evaluation” is more appropriate then “detection”

Line 28-40: please, provide some references

Line 53: please provide what US is.

Methods

Line 84: please write the M is mean and SD is standard deviation

Line 95: please correct the table. n (%) should be written where the number and the percentage are. Please also prove in notes that n corresponds to “number”.

In the table, the socioeconomic status is presented. How it has been assessed? Please provide information about it.

Figure 1: if possible, it is appreciated if figure 1 is written in English. Thank you. Furthermore, if it possible, it should be appreciated to have also a legend / notes to know what T or S is… Finally, the figure has numbers that range from 3 to – 4, I suggest to have values that has the same positive and negative ranges (4 and -4). Furthermore, if printed, both figure 1 and 2, I don’t think they will appear clearly, please provide a better quality of the imagine.

Discussion

Line 255-292: please, justify the findings with some references. Thank you.

Line 323: please, correct: “in his study” in “in this study”

Line 323-331: this part could fit better in the conclusions, not in the limitations of the study

Line 335-336 seems a repetition of line 330-333.

Considering the topic and the aim of the study, I strongly suggest to structure the introduction considering “standard operating procedure” to create guidelines for researchers and teachers in the use of the TGMD-3. From the findings of this study, guidelines could be created and proposed as a standard operating procedure. Interesting studies that can be read to better understand the topic are: Angiuoli SV, et al. Toward an online repository of Standard Operating Procedures (SOPs) for (Meta) genomic annotation. OMICS 12(2):137–141 (2008). https:// doi. org/ 10. 1089/ omi.2008. 0017 ; Petrigna, L. et al. The importance of standard operating procedures in physical fitness assessment: a brief review. Sport Sci Health 18, 21–26 (2022). https://doi.org/10.1007/s11332-021-00849-1

Thank you for the consideration and best regards.

Author Response

(The authors gave the same response as above.)

Round 2

Reviewer 2 Report

Thank you for your consideration for my revisions. The manuscript resulted importantly improved. Consequently, I suggest its publication